# Entrepreneurs and Territorial Diversity: Success and Failure in Andalusia 2007–2015

**Eugenio Cejudo García** *[ID], **José Antonio Cañete Pérez**[ID], **Francisco Navarro Valverde**[ID] and **Noelia Ruiz Moya**

Department of Human Geography, University of Granada, 18071 Granada, Spain; joseaca@ugr.es (J.A.C.P.); favalver@ugr.es (F.N.V.); noeliaruiz@ugr.es (N.R.M.)
* Correspondence: cejudo@ugr.es

**Abstract:** Rural Europe today cannot be understood without considering the impact of the EU's Liaisons Entre Actions de Developpement de l'Economie Rurale (LEADER) rural development programme. Although in general it has had a positive impact, research has also revealed spatial and social disparities in the distribution of funds. Our primary source was the files for all the LEADER projects processed in Andalusia between 2007 and 2015. In addition to successfully executed projects, we also focused on "unfunded" projects, those in which, although promoters had initiated the application procedure, a grant was never ultimately obtained. Project failure must be studied so as to avoid biased findings. We then classified these projects within the different types of rural area and analysed the behaviour of the different promoters in these areas. Relevant findings include: project success or failure varies according to the different types of rural area, as does the behaviour of the different promoters; the degree of rurality can hinder project success; young and female entrepreneurs were more likely to fail; the type of promoter is strongly influenced by the distance to cities in that companies and Individual Entrepreneurs tend to invest in periurban spaces, while public sector promoters such as Local Councils are more prominent in remote rural areas.

**Keywords:** neo-endogenous rural development; LEADER approach; rural areas; classification and types of rural areas; good practices; rural depopulation and aging; young and female entrepreneurs; entrepreneurship; funded and unfunded projects; Andalusia

## 1. Introduction: State of the Art

The current situation of rural areas cannot be fully understood without taking into account the impact of the LEADER programme. LEADER, an acronym for its French title "Liaisons Entre Actions de Developpement de l´Economie Rurale", has been applied throughout the rural areas of the European Union (EU). It was created as a "laboratory" for innovation which could strengthen local capacities and help solve problems in rural areas, via a strongly territorial, "bottom-up" approach. Since it was first established at the beginning of the 1990s, it has become the most emblematic practical application of the recent theories of neo-endogenous rural development on which it is based. The aim of LEADER is to plant the seeds for strong, self-sustaining rural development. The main specificities of this approach are: to promote innovation, above all social innovation; the integrated, multi-sector nature of the projects; the territorial perspective; networking; economic diversification; the bottom-up approach; local decision-making. Originally established as an European Economic Community (EEC) Initiative (1991–2006) implemented through Local Action Groups (LAGs) made up of entrepreneurs, public institutions and civic associations, it was later integrated (since 2007) into the corresponding national and regional Rural Development Programmes, with specific LEADER actions.

Its implementation, with varying degrees of success [1–3], has revealed among other things: the unequal territorial distribution of LEADER funds [4–8]; the development of important social innovation processes in rural areas [9,10]; the varying participation of the different stakeholders as promoters of LEADER projects [11,12]; the vital importance of social capital in rural development processes [13–17]; and the importance for rural development of the promotion and enhancement of natural and cultural heritage, both as cohesive elements of local identity that must be protected and as hugely powerful assets for enhancing rural tourism. These are both emblematic aspects of the LEADER programme. On these lines, various articles have examined the impact of LEADER on for example ways of combining traditional agricultural and livestock practices with agritourism [18,19] wine tourism [20,21], olive oil tourism [22], dehesa grasslands [23], landscapes [24], local skills, knowledge and festivals [25] or the impact on the structure of the rural tourism sector produced by LEADER-related actions, which resulted in an excess supply of accommodation, which was often of poor quality [26]. Other research has focused on the role of LEADER in halting the depopulation of rural areas [27–29], strengthening their level of resilience [30] or simply, as a new methodology for intervention in the development of rural areas [31,32], known as the LEADER approach [33,34].

Both these and many other articles that could be cited centre on LEADER projects that have been successfully carried out and tend to ignore those other projects in which although the promoters had begun the LEADER grant application procedure, a grant was never ultimately obtained. In this article, we will be referring to these projects as unfunded projects. In other words, research on the LEADER programme has tended to focus on funded projects and has largely ignored the projects that applied for but did not finally receive financial support from the programme. We believe, together with Rodríguez et al. [35] (pp. 103–104), that it is also necessary to study issues such as failure, inefficiency and the incapacity to foresee change, so as to avoid biased explanations of social action that tend to marginalise those who do not fit into prevailing success-linked models.

This is why the only research in the literature that deals with the question of unfunded projects, does so indirectly. Dargan and Shucksmith [36] (p. 285) talked about a "project class" made up above all of members of the LAG and well-positioned actors in the public and private sector with substantial financial resources, knowledge and innovation capacity, who control and are well informed about LEADER investments. At the other end of the scale, there are other groups including young people and women, who are less involved even though their projects enjoy certain advantages in the selection and funding process. The authors of [37–40] also made it clear that women are less likely to become rural entrepreneurs, even though they are less afraid of business failure. In spite of this, the LEADER programme has contributed, together with other initiatives, to the creation of new identities and social representations of rural women, which have made them more visible [41] and have enhanced social inclusion in a context in which new socioeconomic and spatial realities are emerging in rural areas of Europe [42,43]. This will lead to the progressive empowerment of women in the personal, family, social and political spheres [44].

Our past research on projects of this kind in Andalusia for the programme period 2000–2006 [45–47] revealed first of all that there was a need to improve management and to update the criteria and the processes for the selection and monitoring of projects. We also found that the number of unfunded projects varies greatly from one territory to the next, a fact which was reflected, in an extreme case, in the considerable number of municipalities in which none of the proposed projects were funded. Another weakness of the LEADER approach was that it did not establish specific measures for areas with low population density to combat the problems arising from depopulation. In general in these areas, neo-endogenous rural development action has not achieved the desired results and at times has even proved unsuitable, missing important opportunities to help reverse depopulation. Finally, the typical profile of the promoters of unfunded projects was that of a young person, and in particular a young woman, who was trying to set up a business. The most common legal forms within which these businesses were established were as self-employed workers, limited companies or business partnerships.

Our proposed field of study is therefore quite original, not only because the subject that we have chosen, namely unfunded projects, has rarely been studied in our field of research, as mentioned above, but also due to the level of detail of the information on which our research is based, the individual files for each project, in a territory like Andalusia, a large region with a population of about 8.5 million people.

In this research our aim is to analyse both the unfunded and the successfully executed projects by looking at the number of projects, and the territories in which they implemented or sought to implement these projects, according to different territorial typologies that enable us to assess and compare their behaviour. Our initial hypotheses are that, on the basis of our previous research studies, the groups with the greatest limitations when it comes to starting a business, including among others individual entrepreneurs and the smallest, most vulnerable companies, will be those least likely to try to set up businesses and most likely to fail. In addition, the participation of the different stakeholders will vary according to the territory in question, with the public sector playing a greater role in less developed areas, and private investors dominating in the areas with more dynamic economies.

## 2. Sources, Methodology and Study Area

The basic source we used was the list of projects for which grant applications were processed (12,855) under the LEADER programme between 2007 and 2015. This information was provided by the Department of Agriculture, Fishing and Rural Development of the Regional Government of Andalusia. For comparison purposes, we have separated the projects into executed projects (6225) and unfunded projects (6630). Unfunded projects were considered to be those which, after a grant application had been made and a file had been opened, were ultimately not executed with LEADER funds. This does not necessarily mean that these projects were never carried out as on occasions the promoters decided to renounce LEADER funds so as to qualify for finance from other programmes.

There are various problems involved in working with this source, especially when analysing unfunded projects: missing information as the forms have numerous uncompleted boxes; countless typing errors, mistakes in the coding of some of the variables, etc. We are therefore working with projects in which the information was often not fully filled in or contained errors, only some of which can be corrected, and in the case of the unfunded projects, which were either never carried out or if they were carried out were done so without LEADER funds.

The types of promoters in this study (as listed below in Table 2) are those described in the source and the analysed variable was the number of funded/unfunded projects.

The results of the statistical analysis were input into a Geographic Information System, ARCGIS 10.6, which produced graphic outputs in the form of vectorial plans that were exported to jpg format. We were unable to perform a qualitative analysis regarding the reasons why the promoters of unfunded projects decided not to continue with them.

Although our analyses were conducted at the municipal scale, they were based on individual files, which means that we only studied those municipalities in which files were opened in relation to applications for LEADER grants. Those projects in which it was not absolutely clear in which municipality the project was intended to be carried out were excluded. The results were then aggregated at the regional level in line with the different types of territory established for Andalusia. Adjacent municipalities of the same type were joined together on the map.

The enormous difficulties inherent in establishing a typology of rural spaces in Spain, or in the OECD in general [48], are due to questions such as the availability and reliability of current and historic sources, the scales with which one decides to work, the variables that are used to establish the different typologies (rural, intermediate or urban) or the thresholds which are set to distinguish between them. The Spanish National Statistics Institute (Instituto Nacional de Estadística) uses the total population as a defining variable, establishing a threshold of up to 2000 inhabitants for rural municipalities and up to 10,000 for medium-sized. Municipalities with over 10,000 inhabitants are regarded as urban. This classification is widely used because of the availability and reliability over time of the data, although

certain doubts have also been raised because of the constant need to increase the thresholds to take into account that a municipality may contain various separate centres of population [49]. However, this typology does not always adapt to the peculiarities of the territorial structure, as happens in our study area, Andalusia, in which the typology adapts poorly to a region in which "agri-towns" [50–52] or intermediate towns [53,54] play a very important role.

Following the recommendations of the OECD [55], the European Union established three large territorial categories (mainly rural regions, intermediate regions and mainly urban regions) on the basis of a benchmark population density figure of 100 inhab/km$^2$ used to distinguish rural municipalities from urban ones. Under this system, the mainly rural regions are those in which over 50% of the population live in rural municipalities; the intermediate regions are those in which between 15% and 50% live in rural municipalities; meanwhile, the mainly urban regions are those in which less than 15% of the regional population live in rural municipalities. This classification could be applicable to NUTS 3 regions. In recent years, interesting proposals have emerged in this regard at the local level. Firstly, Molinero [56] established a rural typology in which population density was the main criterion. This is because population density is a key factor in any rural development policy and since the 1990s has been the most frequently used criterion by the OECD, the EU and the Spanish Ministry of Agriculture, as well as by geographers and territorial planners. This classification developed from Law 45/2007 on the Sustainable Development of Rural Areas promoted by the Ministry of Agriculture, which classified as rural all those municipalities with less than 30,000 inhabitants and less than 100 inhabitants/km$^2$. This group was then subdivided into three types of rural municipality: deep < 5 inhab/km$^2$; stagnant "5" to "24.9"; and dynamic "25" to "99.9" inhab/km$^2$. The application of this classification in rural spaces in Andalusia could be problematic due, as mentioned earlier, to the socio-territorial importance of agri-towns in the region.

Secondly, de Cos and Reques (2019) [57] proposed a typology of spaces based on their territorial and demographic vulnerability using cartographic sources available in GIS format, taking advantage of new European and Spanish legislation enabling access to official cartographic databases in digital format. For this typology, a multi-criterion analysis involving a weighted linear combination was applied. Although the methodology and the resources used appeared to us to involve a very important qualitative leap in an attempt to go beyond classifications based on population or density, the resulting aggregation of results in nine categories according to the degree of vulnerability would be difficult to apply in this research study. In addition, while the notion of territorial vulnerability fits quite well with the real situation in Andalusia, that of demographic vulnerability does not provide such a good fit.

One of the most widely cited proposals for the classification of rural areas in Spain was presented by Reig, Goerlich and Cantarino (2016) [58]. These authors based their proposal on the classification made by the OECD and the EU, which was itself based on population variables such as density, and took the 1 km$^2$ grid as a spatial reference for analysis. The use of newly available georeferenced data as to exactly where each inhabitant lived within the municipality enabled them to avoid all the distortions caused by calculating the population density on the basis of the total area of the different municipalities, in which there are often large areas with little or no population. They also included, in line with other research work being conducted in the EU, accessibility to urban centres and to services, considering for this purpose the closest towns or cities with a population of over 50,000 people [59]. They also looked at land uses in order to classify intermediate and urban areas into closed and open spaces, and used the time taken to access services to classify rural territories into near (up to 45 min) and remote (more than 45 min). On the basis of this classification and taking into account that our analysis focuses above all on rural areas in that it examines projects linked to the LEADER programme, we decided to modify this classification system, applying as a discriminatory variable the time taken to access services. On this basis, the intermediate municipalities were divided into near and remote, depending on whether or not they were over 30 min from a city (as most are situated in parts of the Guadalquivir Valley with a high population density). In rural areas, a third category was established due to the widely diverse range of situations observed in the different municipalities. These were divided into "near"—those

less than 45 min away from a city—, "remote"—between 45 and 60 min—, and "deep"—60 or more min away—(Figure 1). We believe that with the aforementioned modifications, this is the classification that best adapts to the real situation in Andalusia.

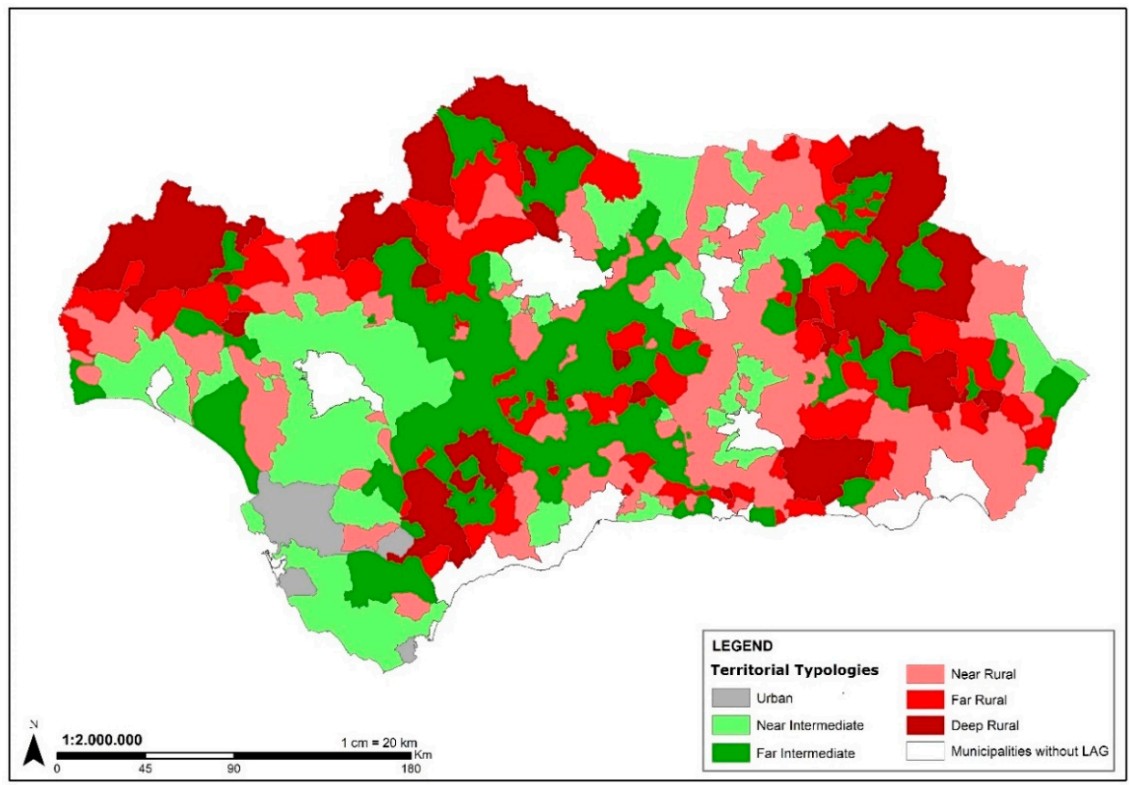

**Figure 1.** Territorial typology of Andalusia (Reig et al.) [58], adapted by the authors.

Table 1 presents various synthetic indicators of sociodemographic aspects of the different typologies. The table was drawn up using data from 2011. This year was chosen as a reference because it falls halfway through the study period (2007–2015) and because census and local registration information is readily available.

After a brief analysis of the data presented, we found that in 2011, 5.9% of the municipalities in Andalusia were urban areas. These covered 6.7% of the total surface area and were home to almost 50% of the population, with very high densities. The population of these municipalities continued to rise over the study period, increasing by 168,940 inhabitants, 48.3% of the total increase across the region. This trend continues the pattern which first appeared in Spain in the 1960s as witnessed by the fact that the population in these municipalities rose by over 40% between 1961 and 2011. These municipalities are generally situated in flat areas at an average altitude of less than 260 m and are very close to areas that provide services at a distance of just 4 min. They have the lowest average age population and a relatively high proportion of the population are over 65. The agricultural sector is relatively insignificant, as can be seen from the number of people affiliated to the agrarian section of the Social Security system, who account for less than 4% of the population in open urban areas.

By contrast, 66.1% of the municipalities are classified as rural. These cover 52.7% of the territory and house 12% of the population (2011). In demographic terms, over the period 2007–2015 the population of these municipalities fell by 1825 people, although the greatest losses were in the regions furthest away from service centres (Regions 6 and 7, Types 6 and 7 of Table 1), and in fact there were gains in the nearest areas (Region 5, Table 1), although these were not sufficient to make up for the losses in the more remote areas. A trend shared by all three types of rural municipality was that their population in 2011 was less than that in 1961 with accumulated losses of over 22.3%. The decline was more intense the more remote the municipality, as can be seen by the fact that almost half the loss of

population took place in Category 7 areas (Table 1). The rural municipalities are normally situated at higher altitudes of between 518 and 718 metres on average and tend to be further away (between 30 and 70 min) from the services provided by towns and cities with populations of over 50,000 inhabitants. These rural municipalities also have the oldest populations with an average age of 46 years old and well over 20% of the total population aged 65 years old or over. Perhaps the most serious statistic in the villages in Category 7 (Table 1) is the aging rate (the ratio between people aged 65 or over and people aged under 15), which is twice the regional average (183 compared to the regional average of 93). As might be expected, the figures for affiliation to the Social Security system clearly reflect the continued dominance of the farming sector, as can also be seen from the number of people claiming the Agricultural Unemployment Subsidy and the Agrarian Income Supplement, benefits received by temporary farmworkers in Andalusia and Extremadura [60–62].

Finally, and so as not to extend this territorial presentation unnecessarily, the intermediate municipalities accounted for 27.9% of the total, 40.6% of the surface area and 38.4% of the population. In general, the variables for the intermediate municipalities range between the other two categories, although we should also highlight Category 4, Remote Intermediate, a category which normally coincides with the agri-towns, located at some distance from the services provided by the city (on average 45 min away). These towns act as capitals of their respective subregions and perform a key function in the provision of basic services and facilities, both public and private, that are highly essential in rural Andalusia.

**Table 1.** Socioeconomic indicators according to territorial typologies.

| | Urban | | Intermediate | | Rural | | | Total |
|---|---|---|---|---|---|---|---|---|
| | Closed | Open | Near | Remote | Near | Remote | Deep | |
| | Type_1 | Type_2 | Type_3 | Type_4 | Type_5 | Type_6 | Type_7 | |
| Nº municipalities | 28 | 18 | 125 | 91 | 228 | 132 | 151 | 773 |
| Area km$^2$ | 826 | 5071 | 16,500 | 19,170 | 18,895 | 12,138 | 15,186 | 87,786 |
| Population_2011 | 2,086,485 | 2,071,715 | 1,997,533 | 1,225,614 | 555,114 | 238,687 | 214,696 | 8,389,844 |
| Density_2011 | 2.527 | 409 | 121 | 64 | 29 | 20 | 14 | 96 |
| Travel time | 3.8 | 3.4 | 17.3 | 45.0 | 30.3 | 51.7 | 73.3 | 40.4 |
| Altitude | 256 | 264 | 267 | 381 | 518 | 631 | 718 | 504 |
| Pop_Women | 51.7 | 51.3 | 49.4 | 50.1 | 49.0 | 49.3 | 49.3 | 50.5 |
| Pop_Men | 48.3 | 48.7 | 50.6 | 49.9 | 51.0 | 50.7 | 50.7 | 49.5 |
| Pop/Municipality | 74,517 | 115,095 | 15,980 | 13,468 | 2435 | 1808 | 1422 | 10,854 |
| Pop. Growth_2007–2015 | 88,402 | 80,538 | 182,781 | 2659 | 20,825 | −8123 | −14,527 | 352,555 |
| Pop. Growth_2011–1961 | 43.3 | 41.9 | 36.7 | 1.3 | −6.6 | −7.1 | −9.6 | 100.0 |
| Pop._0–14 years | 15.9 | 16.8 | 17.8 | 16.0 | 14.5 | 12.8 | 12.6 | 16.3 |
| Pop._15–64 years | 68.9 | 68.9 | 69.6 | 67.5 | 66.7 | 65.4 | 64.2 | 68.5 |
| Pop._≥ 65 years | 15.2 | 14.3 | 12.6 | 16.5 | 18.9 | 21.8 | 23.2 | 15.2 |
| Aging rate | 95.7 | 85.4 | 71.1 | 103.3 | 130.3 | 171.0 | 183.3 | 93.3 |
| Average age | 37.9 | 37.7 | 38.1 | 40.4 | 43.3 | 45.7 | 46.0 | 42.7 |
| Affil. General Reg. | 81.0 | 78.8 | 50.3 | 40.8 | 30.0 | 29.8 | 28.3 | 62.7 |
| Affil_Agrarian Reg. | 1.2 | 3.6 | 26.5 | 37.4 | 51.1 | 50.3 | 49.2 | 17.7 |

Source: Instituto de Estadística de Andalucía. The authors.

## 3. Results

It is important to remember that we only included those projects in which a particular municipality was mentioned as the place where the project was to be carried out. All projects without a specific location were excluded as were those that were intended to be executed at the sub-regional rather than the municipal scale. This explains why although 12,855 projects commenced the application procedure, in this study we only analysed 11,104 or 86.4% of the total. Of the 1751 projects that were not included in our study, 1271 were unfunded projects and 480 were successfully executed. Another interesting statistic is that 94.6% of the funded projects were promoted by associations including the Local Action Groups (LAG) (This category includes promoter types G, G14 and J (this classification is followed in Tables 2 and 3).). Associations were also the body that initiated the largest number of unfunded

projects at 427, or 33.6% of the total. This was followed by Individual Entrepreneurs with 365 unfunded projects and various different types of private companies (This category includes promoter types A, B, E and F (this classification is followed in Tables 2 and 3).) with 324. Non-profit making associations such as LAGs were therefore the promoter most affected by our decision to analyse the projects at a municipal scale and are therefore somewhat underrepresented in our results. This is because a lot of the projects presented by these kinds of associations were organised at a sub-regional level rather than a municipal level. This under-recording is substantially less significant in the other variables analysed, although it should also be taken into account. Lastly, the results will be discussed on the basis of five large categories of promoters: Private companies (see note 2); Non-profit making associations (see note 1); Local Councils (code P); Individual Entrepreneurs (code PF) and others (This category includes promoter types Q, R, S and U (this classification is followed in Tables 2 and 3)). Later, we will be looking at some of the components of these large categories in more detail.

### 3.1. Funded and Unfunded Projects. An Overview

The first variable to analyse was the number of projects in which the application procedure for a LEADER grant was initiated. This was done by the type of promoter and by the type of territory, as established above. The initial objective was to answer the following questions: Do the different kinds of promoter act in the same way? Do participation levels vary from one type of territory to the next? Do the different types of promoter have the same probability of success or failure at the outset of the project? Does this vary according to the territory in which the project is to be carried out? In order to help answer these questions, we created Table 2, which contains the data referring to all the projects initiated and Table 3, which shows the ratios between funded and unfunded projects according to the promoter and territory.

**Table 2.** Total number of funded and unfunded projects.

| Promoter | Type 1 | Type 2 | Type 3 | Type 4 | Type 5 | Type 6 | Type 7 | Total |
|---|---|---|---|---|---|---|---|---|
| A | 0 | 11 | 51 | 44 | 24 | 4 | 13 | 147 |
| B | 1 | 64 | 792 | 680 | 439 | 292 | 296 | 2564 |
| E | 0 | 7 | 37 | 50 | 27 | 18 | 16 | 155 |
| F | 1 | 25 | 133 | 122 | 132 | 76 | 69 | 558 |
| *Private companies* | *2* | *107* | *1013* | *896* | *622* | *390* | *394* | *3424* |
| G | 1 | 41 | 203 | 259 | 105 | 84 | 80 | 773 |
| G14 | 0 | 1 | 71 | 100 | 128 | 21 | 36 | 357 |
| J | 0 | 4 | 62 | 32 | 34 | 14 | 9 | 155 |
| *Associations* | *1* | *46* | *336* | *391* | *267* | *119* | *125* | *1285* |
| *P* | *1* | *29* | *472* | *541* | *842* | *511* | *570* | *2966* |
| *PF* | *6* | *138* | *943* | *774* | *655* | *392* | *394* | *3302* |
| Q | 0 | 1 | 6 | 20 | 6 | 0 | 3 | 36 |
| R | 0 | 2 | 22 | 28 | 11 | 13 | 12 | 88 |
| S | 0 | 0 | 0 | 1 | 0 | 0 | 0 | 1 |
| U | 0 | 0 | 1 | 0 | 0 | 0 | 1 | 2 |
| **Total** | **10** | **323** | **2793** | **2651** | **2403** | **1425** | **1499** | **11,104** |

A. PLCs, B. Limited Companies, E. Business Partnerships, F. Cooperatives, G. Associations and Foundations, G14. LAGs; J. Civil Societies, P. Local Councils, PF. Individual entrepreneurs, Q. Public Bodies. R. Religious Congregations and Institutions, S. Departments of Central and Regional Governments, U. Others. Source: Junta de Andalucía. Consejería de Agricultura, Pesca y Desarrollo Rural. The authors.

As regards the number of projects commenced (Table 3), we observed that these were shared out at roughly a third each between three main promoters: Private Companies, 30.8%, Individual Entrepreneurs 29.7% and Local Councils 26.7%. The "Others" category was almost irrelevant at 1.1%, while that of Associations came to 11.6%; although as mentioned earlier, this category was clearly underrepresented. Within private companies, limited companies, often regarded as the poor relations within this group, play a central role as they are responsible for initiating the highest number of projects

with 23.1% of the total. Another trend worth noting was the increasing importance of Cooperatives, although this was less obvious in terms of the number of projects, in which they accounted for just 5%.

If we take the above information about all 11,104 projects and we break it down into executed and unfunded projects, can any differences be observed in terms of the way the different promoters acted in the different territories? In order to answer this question, we have drawn up Table 3, which shows the ratio between funded and unfunded products multiplied by 100 so as to make it easier to understand.

**Table 3.** Ratio of funded to unfunded projects.

| Promoter | Type 1 | Type 2 | Type 3 | Type 4 | Type 5 | Type 6 | Type 7 | Total |
|---|---|---|---|---|---|---|---|---|
| A | 0 | 57 | 82 | 132 | 140 | 0 | 86 | 96 |
| B | 0 | 83 | 99 | 86 | 101 | 67 | 74 | 88 |
| E | 0 | 133 | 147 | 138 | 125 | 100 | 220 | 138 |
| F | 0 | 150 | 217 | 213 | 238 | 145 | 156 | 195 |
| *Private companies* | *0* | *95* | *110* | *101* | *123* | *78* | *89* | *102* |
| G | 0 | 71 | 81 | 106 | 98 | 83 | 111 | 93 |
| G14 | 0 | 0 | 407 | 335 | 191 | 950 | 260 | 280 |
| J | 0 | 100 | 100 | 52 | 70 | 100 | 80 | 80 |
| *Associations* | *0* | *68* | *117* | *141* | *140* | *119* | *142* | *129* |
| *P* | *0* | *21* | *106* | *103* | *190* | *134* | *118* | *130* |
| *PF* | *100* | *106* | *89* | *85* | *85* | *102* | *91* | *90* |
| Q | 0 | 0 | 20 | 67 | 100 | 0 | 50 | 57 |
| R | 0 | 100 | 144 | 300 | 175 | 117 | 140 | 175 |
| S | 0 | 0 | 0 | 0 | 0 | 0 | 0 | 0 |
| U | 0 | 0 | 0 | 0 | 0 | 0 | 0 | 0 |
| **Total** | **67** | **85** | **102** | **101** | **130** | **106** | **104** | **107** |

A. PLCs, B. Limited Companies, E. Business Partnerships, F. Cooperatives, G. Associations and Foundations, G14. LAGs; J. Civil Societies, P. Local Councils, PF. Individual Entrepreneurs, Q. Public Bodies. R. Religious Congregations and Institutions, S. Departments of Central and Regional Governments, U. Others. Source: Junta de Andalucía. Consejería de Agricultura, Pesca y Desarrollo Rural. The authors.

For Andalusia as a whole this ratio is 107, which means that slightly more projects were implemented than were not. When these values are analysed by a promoter, important differences emerge. The promoters that achieved above average rates of implementation and could therefore be considered as being better funded are Cooperatives, LAGs and Local Councils (P). The difference between these groups is important in quantitative terms. Cooperatives and LAGs obtained ratios that were double the average value while the ratio for Local Councils was 21% above average. At the opposite end of the scale, in which there were more unfunded projects than funded ones, the Individual Entrepreneurs and Limited Companies stood out with 17 and 19 percentage points less than the average for Andalusia, respectively. These are private investors, who are vitally important in terms of the number of projects they promoted and above all in terms of the amounts invested and of the associated employment. They are also the ones that take the biggest risks in terms of investment as they are investing their own capital and because they receive proportionally smaller grants compared for example to Local Councils and LAGs.

If we carry out a more in-depth analysis of the behaviour of the promoters according to the different types of territory, various interesting questions come to light. The success/failure ratios for Individual Entrepreneurs were below the regional average of 107 in all the different types of territory, a very clear sign of the weakness of this group when it comes to implementing a project. Their highest levels of failure were located in the intermediate regions, especially Remote Intermediate areas, and in Near Rural areas. These areas were also the ones in which most projects were started. The ratios were higher at the extremes, in particular in Remote and Deep Rural areas in which the fact that there was a small number of projects and of promoters seemed to help more solid business proposals to come to fruition. At the opposite end of the scale in Urban areas, the higher ratio was due to the more dynamic economic environment and to the fact that a relatively small number of projects (144) were

commenced. The other large category in which there was a majority of unfunded projects was in private companies, in which there were important internal differences as mentioned earlier. If we look at private companies in general, we observe that the most important differences in their results are due more to remoteness/nearness than to the distinction between rural, intermediate and urban areas. The Near Intermediate municipalities obtained a score of 110 compared to 101 for the remote areas, while in the rural areas the maximum value was 123 for the Near Rural municipalities compared to 78 and 89 for the Remote and Deep Rural areas, respectively. Within this category, limited companies started by far the highest number of projects. They established a general trend but with lower values in all the different types of territories, such that they only surpassed a ratio of 100 in Near Rural areas and even then by very little (101). For their part, PLCs had high levels of success in the execution of their projects in Remote Intermediate and Near Rural areas with 134 and 140, respectively, while their scores were over 50 points lower in all the other types of territory.

In three types of promoters, the number of funded projects was far in excess of that of unfunded projects. These included Cooperatives, linked above all to the farming sector, in which there were twice as many funded projects as unfunded projects with values that were much higher in intermediate and Near Rural areas than the already high average for this category of 195. In Remote and Deep Rural areas the scores were below the average for Cooperatives but were still 40 points above the regional average for all projects (107). The average value for the LAGs was almost 3 times the regional average of 107 and varied enormously between the different types of territories, something which can be explained in part by the small number of projects initiated. In addition, many of their projects were only activated at the end of the programming period, on quite a number of occasions so as to make up for the absence of other promoters by turning to a "reserve stock" of solidly constructed projects for which finance was assured. Lastly, Local Councils showed their highest levels of success in all three types of rural area, reaching their maximum in Near Rural in which there were almost twice as many funded as unfunded projects. This value was notably lower in Remote Rural areas (134) and Deep Rural areas (118), and far lower in the intermediate regions, at just over 100.

In summary, for most of the actors involved, the remoteness and the degree of rurality of the municipalities proved a handicap that made it more difficult for the projects commenced under the aegis of the LEADER programme to be successfully executed; the exception to this rule was Individual Entrepreneurs, an important finding that must be borne in mind.

## 3.2. Geographical Distribution across Andalusia of the Different Types of Area

As can be seen in Figure 1, the classification of rural spaces in Andalusia according to the nomenclature proposed by Reig et al. (2016) [58] adapts quite accurately to a territorial structure in which the mountain areas are quite different from those situated in the valleys. The eastern side of the region is dominated by rural areas (Near, Remote and Deep), in sharp contrast to the flat plain traversed by the River Guadalquivir, which is dominated by intermediate regions and even a few urban areas. The latter are mostly situated around the Cádiz metropolitan area and Algeciras.

By contrast, the most strongly rural areas (in their different categories) can be seen in practically all of Sierra Morena, with the exception of a few slightly larger municipalities in the Valle de los Pedroches and Andújar. The rural area covered by the Baetic and Sub-Baetic Cordilleras is also easily distinguishable because it dominates the eastern half of Andalusia.

Calculating the ratio between funded projects and unfunded projects is a way of assessing how effectively the LEADER projects have been managed. The results set out in Figure 2 in relation to Individual Entrepreneurs as promoters can only be described as "disappointing". In practically all types of territories and regardless of their geographic location, there were more unfunded projects (those initiated and processed but ultimately never executed) than funded or executed projects. An even balance between unfunded and funded projects was only observed in Remote Rural areas, in which the ratio was around one, and in the areas classified as Urban, in which it was 1.06.

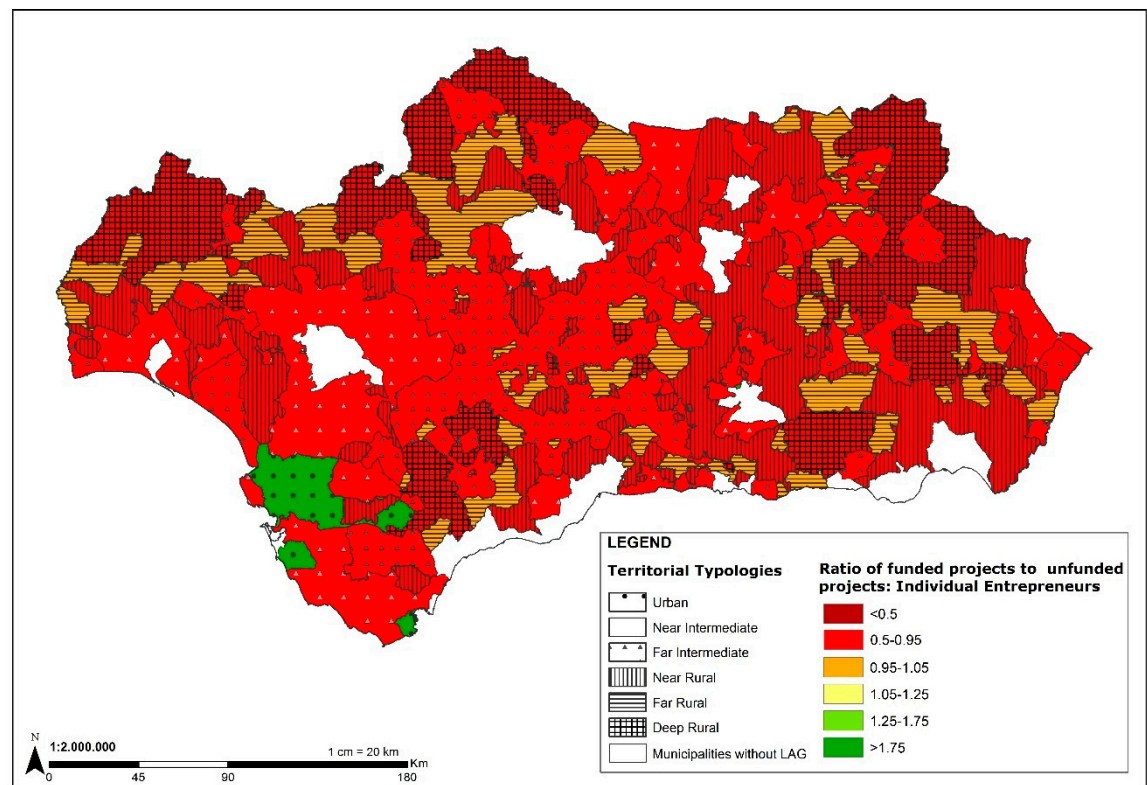

**Figure 2.** Ratio between the number of funded/unfunded projects initiated by Individual Entrepreneurs according to territorial typology. Source: Junta de Andalucía. Consejería de Agricultura, Pesca y Desarrollo Rural. The authors.

This means that in all the territories, regardless of their degree of "rurality", it was self-employed promoters (and within them young people and women), who found it most difficult to implement their projects due to administrative problems, lack of finance, etc.

Figure 3, which refers to public promoters, highlights a completely different situation to that described above. Town Councils promoted more funded projects than unfunded ones. In Near Rural areas the former almost doubled the latter, while in Remote Rural areas, the difference was slightly lower. However, and this is very significant, very similar ratios were observed in Deep Rural areas, often the most depressed regions with worse social and territorial conditions for the funded establishment of private businesses. In these areas in which public investment is urgently required, the proportion of unfunded and funded projects was very similar, as happened in the Near Intermediate areas. In Urban areas there were more unfunded projects than funded ones. This was followed by Remote Intermediate areas, although in the latter the ratio values were very close to 1.

The behaviour of private companies (Figure 4) is clearly associated with the degree of "rurality" of the area in question. The more rural the area is, the higher the proportion of unfunded projects. For Andalusia as a whole, the ratios vary from 0.78 in Remote Rural areas to 1.23 in Near Rural. This confirms once again that proximity to cities is an important factor in the success of LEADER projects. Similarly, in remote inaccessible areas it seems more difficult to bring projects to funded fruition. This map highlights once again the differences between the Guadalquivir Valley and the mountainous areas of Andalusia.

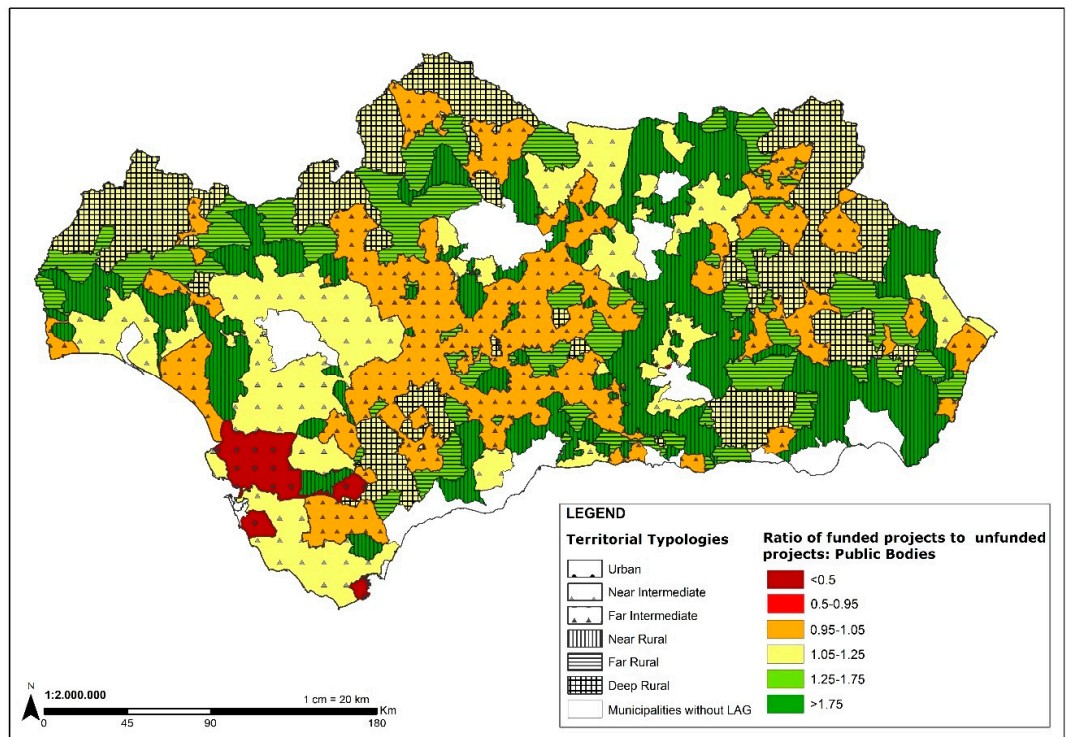

**Figure 3.** Ratio between the number of funded/unfunded projects initiated by Public Bodies according to territorial typology. Source: Junta de Andalucía. Consejería de Agricultura, Pesca y Desarrollo Rural. The authors.

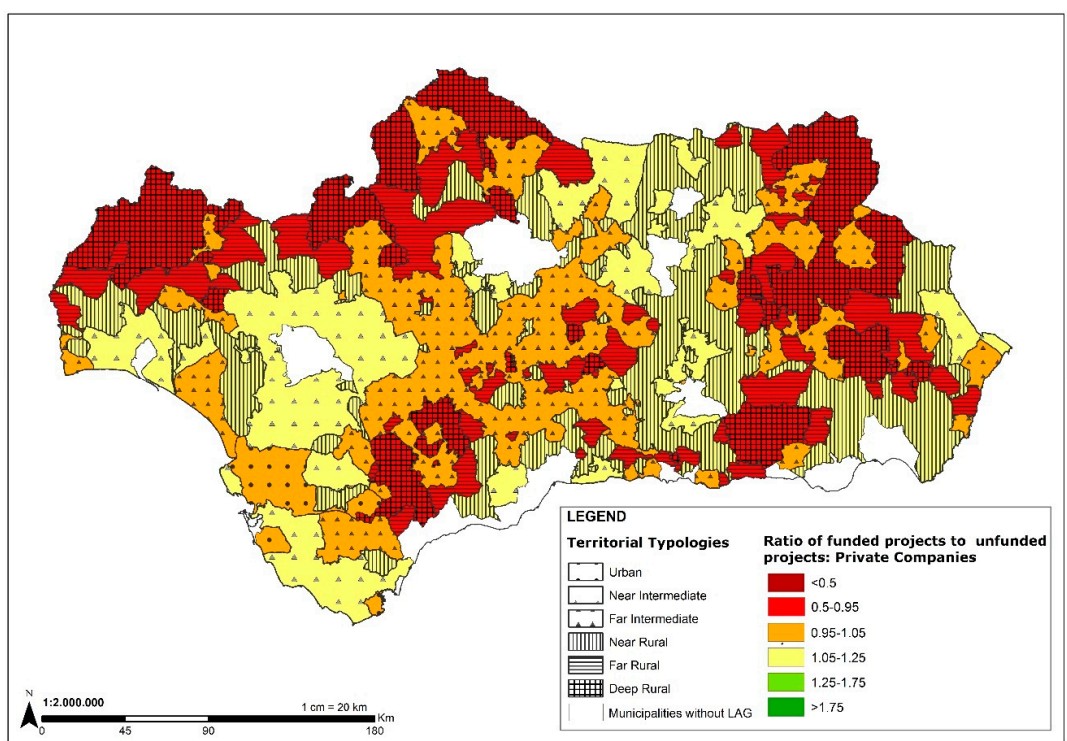

**Figure 4.** Ratio between the number of funded/unfunded projects initiated by private companies according to territorial typology. Source: Junta de Andalucía. Consejería de Agricultura, Pesca y Desarrollo Rural. The authors.

## 4. Discussion and Conclusions

This study, which analyses all LEADER projects for the period 2007–2015, both funded [63] and unfunded, taking projects at the municipal level as a reference, has confirmed previous findings that the participation of the different promoters of LEADER projects varied greatly within the Andalusia region. In addition, and this is the most novel aspect of our research, their intervention varied according to the type of project and the different types of area established. Our results show that the remoteness and rurality of some of the territories made it more difficult for most promoters to successfully conclude the projects they initiated under the LEADER programme, although not all were affected in the same way or to the same degree. Our findings also confirmed that nearness to cities also has a strong influence on country areas in terms of economic activity and income [64]. This is especially evident amongst private sector actors such as companies and Individual Entrepreneurs, who tend to invest in these areas, while the opposite is true for public sector promoters such as Local Councils.

Another important conclusion ratified in this research is the territorial complexity of Andalusia, which is difficult to fit into any general territorial classification system designed for Spain as a whole. This is due above all to the presence of large numbers of medium-sized towns with a strong rural/farming component, known as "agri-towns", so confirming the position defended by Sánchez [54] (p. 189) who argued that these towns are "first and foremost, an opportunity for the territorial development of Andalusia" because they strengthen the hierarchical, balanced structure required for regional development due to their broad spatial distribution and their enormous functional and social diversity, which results in spaces that are highly favourable for business investment and offer a high quality of life for local residents whose numbers continue to grow, so reinforcing the trends that favour the flat areas compared to the mountains, the large compared to the small and the coast compared to inland regions [65,66]. In spite of this, the classification proposed by Reig et al. [58], which we have slightly amended, adapts well to the territorial structure of Andalusia, which is clearly marked by the divide between the mountain areas in the Eastern half in which there are a majority of rural areas with relatively small villages and the flat plain dominated by the Guadalquivir Valley, where most of the intermediate areas, many of which are agri-towns, are located (in general the coastal areas of Andalusia are not considered rural and are not covered by the LEADER programme).

The execution of the LEADER programme (2007–2015) was affected by the economic and financial crisis that erupted in 2008 and continued throughout the programme period. This resulted in a final investment in Andalusia of 514.1 million euros and a subsidy of 209.1 million, a mere 55.4% and 60.2% of the amounts spent during the previous programme period (2000–2006). Likewise, the total number of projects was only 75.8% of those carried out in the previous period. The average investment per project of almost 82,600 euros was also 27% lower.

The difficulties faced by both public and private investors resulted in constant changes in the National Strategic Plan (PEN) and in the different Rural Development Programmes (there were 10 different versions in Andalusia). Some of these changes were forced upon them by changes in European legislation or due to alterations in LEADER Axes 3 and 4 in which the EAFRD funds initially allocated to LEADER (10%) were reduced to the new minimum of 5% established by the EU in 2012 [67]. These issues were also noted by the Court of Auditors of the European Union in its 2010 report [68] on the implementation of LEADER at the beginning of the mainstreaming period. The economic crisis also damaged the capacity of the welfare state to combat poverty and inequality. This had serious effects in Mediterranean areas, which contain some of the most vulnerable social groups and territories in Europe [69]. The austerity conditions imposed on the most affected countries, Spain included, and the preference at European level for flexibility in the labour market, referred to as "flexisecurity", made businesspeople vulnerable to economic flows. At the same time, workers had the moral duty to empower themselves by acquiring the capacity to adapt [70], which, depending on a series of contextual and individual factors, led many salaried workers to become "entrepreneurs out of need" [71].

Our research has also highlighted the importance of PLCs and of limited companies when it comes to promoting rural development. In Spain, limited companies can be set up with less initial

share capital (€3000) than PLCs which require a minimum share capital of €60,000, €15,000 of which must be paid up on incorporation of the company; the bureaucratic procedure required to set up a limited company is more flexible, enabling a more family-based ownership structure with relatively few shareholders. Limited Companies are therefore the type of company that best adapts to the socioeconomic reality of intermediate and rural areas. However, for these same reasons they are more likely to fail than PLCs. It could therefore be argued that the patterns observed over the period 2000–2006 have been repeated [12]. These companies have a much greater presence in intermediate areas and in Near and Remote Rural territories, and are less evident in Deep Rural areas. They take advantage of the dynamism associated with urban areas, but the fact that they are easy to set up and do not require much stock capital means that they also top the bill in terms of investment and employment in all the different types of territory.

The role of Cooperatives is also worth highlighting. Firstly, because their results for several variables meant that they were second in importance within the group of private companies, a long way ahead of PLCs. As regards the level of success of the projects they started (as measured by the funded/unfunded projects ratio), Cooperatives came second only to LAGs, the promoters with the highest success ratio, which indicates the firm, solid grounding of their business proposals. Secondly, because of the social, mutually beneficial intentions of these ventures, which enhance the activation and consolidation of social capital, an essential feature of rural development processes [16,72,73]. Finally, because this is evidence of the crucial role in rural development that the modernization and enhanced competitiveness of the farming and agro-industrial sectors have been acquiring since the programme period of 2007–2013 [74]. This has also been reflected in international trade and in key sectors at the national and Andalusian levels, such as fruit, vegetables and vegetable oils—and in particular olive oil—[75] and even in innovation in the rural world [76]. This is manifested for example by the fact that the Gross Value Added (GVA) of the farming sector in Andalusia in 2018 represented 5.9% of GDP, compared to 2.5% in Spain and 1.1% in the EU 28, respectively, and 31.4% of the GVA produced by the farming sector in all of Spain. In addition, the value of agro-industrial production in Andalusia accounted for 25.7% of the total for the industrial sector in the region, which is five percentage points higher than the national figure for Spain as a whole. Similar patterns can be observed in farming employment, which at 8.3% of the total was twice the national and EU28 average, while Andalusia's agro-industry accounted for 24.3% of jobs in the region's industrial sector (Junta de Andalucía, 2019). All these statistics highlight the strong territorial, essentially rural implementation of these sectors. In short, the investments linked to the farming and agro-industrial sector (Measure 411 of the LEADER axes) carried out by Limited Companies, Cooperatives and to a lesser extent Individual Entrepreneurs have proved to be a key factor in rural development in Andalusia over the period 2007–2013, above all due to their strong presence in the inland and mountainous areas of the Penibaetic and Sub-Baetic Cordilleras and at the expense of the Guadalquivir Valley [63].

The dynamizing and rebalancing role that should be played by the LAGs through their initiatives, although very limited by the rules applied during this programme period, was almost irrelevant in the least dynamic areas that most required this kind of intervention. This confirms questions that have already been raised such as the increasingly bureaucratic procedures and the very limited citizen participation in these bodies [77–79], the shortages and frequent turnover of staff, as well as the interference from regional government bodies in the performance of their functions [80], so restricting one of their basic principles, namely subsidiarity [26]. All the above does not release the LAGs themselves from their share of responsibility especially as regards greater inclusion within their decision-making bodies (the General Assembly and the Governing Board) of underrepresented groups such as women and young people [37] and of production sectors such as the farming and agro-industrial sector (Matthews, 2005), which can contribute to the dynamizing role that the LAGs have traditionally performed [81]. It is also important to remember the administrative instability that various LAGs in Andalusia have experienced during this programme period, in which two

LAGs have been wound up (Ronda-Málaga and Almanzora-Almería) and the manager of a third (Apromontes-Granada) has been accused in criminal proceedings.

The crucial role played by public sector actors in rural development is undeniable, especially in rural territories and above all in Remote and Deep Rural areas. However, these public sector players will not be sufficient by themselves to revive the fortunes of these territories. Local Councils, although poorly equipped in terms of economic and human resources, can have an enormous impact on the quality of life in their towns and villages in the sense that they have direct, in-depth knowledge and a comprehensive, overall view of the problems in their communities. On election, most take on a commitment to act to resolve these issues, which at least potentially could make them agents or catalysts for innovation, especially in small and medium-sized municipalities [82,83].

Small municipalities have a potential for innovation that many do not fully materialise. These opportunities include for example soft and intangible innovations, the wellbeing of local communities, skills development for local people, smart specialization strategies, bio-economy, eco-economy, social and cultural innovation, community projects, a territorial approach, linkages between agriculture and the wider economy and the promotion of natural resources [84]. In theory, small municipalities are suitable spaces for innovation but this potential is often frustrated by the very limited capacity of the engines that drive innovation. The end results in terms of innovation are very modest. Improvements could be made by recognising that we are interdependent and extending and enhancing networks based on relationships, exchanges and dialogues that foster ideas and learning; it is also vital to improve local leadership that is capable of bringing together and listening to the different stakeholders and generating synergies between them, a situation in which Local Councils or LAGs could act as bridges between people to multiply ideas and create innovation. Finally, it is essential to encourage a feeling of community, so helping create a more cohesive society that is open to people from outside [85].

Small local councils must assume a key role in the development process, focusing local strategies on discovery rather than on individual innovations. They must also offer their own vision about the particular form of development to be pursued, as to how the economic structure should evolve and the changes required to open up the economy to a new field of opportunities. It is clear that no single municipal government can manage the global challenges of aging population, unemployment and social inequalities by itself. Interaction with other tiers of government must therefore be taken into account when designing a governance structure for local policy. The improvement of public service delivery and the creation of multifunctional and mobile services must also be priority objectives [86]. Finally, institutional support must be given to rural development initiatives and possible strategic alignments must be sought between local, regional, national and supra-national policy agendas, with a view to developing a range of complementary policies [84].

While the participation in rural development of the actors mentioned above is important, the participation of individual entrepreneurs is absolutely essential. As private agents of development, they are more often to be found in the most dynamic areas which have the greatest, most certain investment opportunities. However, we believe that the important thing is their constant presence in rural areas with near average or above average values, even in Remote and Deep Rural areas. A fact that should be emphasised given that these areas are the most vulnerable, least dynamic and generally most neglected by promoters of LEADER projects [4,6,87]. They are also areas in which the population is not only poorer but feels poorer, a fact that highlights the need for territorial policies to take into account the heterogeneous nature of municipalities [88] in the design of these policies in which a greater role must be given to the variables of economic geography [89].

Although it was beyond the scope of this analysis, other recent research studies point to the fact that in addition to the typical profile of a mature woman with a low level of training/education, who is running a family business and has family responsibilities and loyalties that can impinge upon business performance [90] and of the "entrepreneur out of need" to whom we referred earlier, new forms of women entrepreneurs are emerging with links to professional services and rural tourism [91]. These combine with a generation of highly trained young women who have returned to rural areas of

Spain with good communication infrastructures, which they see as suitable places for production and innovation in an effort to halt or mitigate depopulation [92].

However, these encouraging signs should not make us overlook the fact that women and young people are the groups that benefit least from these initiatives. Firstly, they have to overcome a large number of obstacles when trying to start up new businesses within the LEADER programme [93]: they carry out far less projects than their adult, male counterparts; they receive smaller average grants per project and the grants they receive make up a smaller percentage of the total amounts invested. They also have a much lower ratio of funded/unfunded projects. In addition, the traditional division of gender roles remains strong in many rural areas, such that women continue to bear the burden of housework and childcare even when they are the only breadwinners in the family unit [94,95]. These traditional gender roles also tend to channel the projects proposed by women investment into sectors such as tourism, food, social care services and handicraft-related activities. A final, very serious issue in Spanish society today is that the increasingly precarious job market and salary system are no longer the exception and have now become the rule for the majority of the population, especially if you are young and/or a woman [96], a fact that is often reflected in LEADER projects, in which precarious jobs tend to be held by women and young people. These questions in relation to depopulation, women, gender and young entrepreneurs need to be addressed in more extensive future research, in which each issue can be analysed separately.

**Author Contributions:** Conceptualization, E.C.G. and F.N.V.; methodology, J.A.C.P.; software, N.R.M.; investigation, E.C.G., F.N.V. and J.A.C.P.; writing—original draft preparation, E.C.G., F.N.V. and J.A.C.P.; writing—review and editing, E.C.G. and F.N.V.; Visualisation, N.R.M. All authors have read and agreed to the published version of the manuscript.

**Funding:** This study was carried out as part of the research project entitled "Successes and failures in the practice of neo-endogenous rural development in the European Union (1991–2014)" funded by the Spanish Ministry of Economy and Competitiveness within its Excellence Programme, CSO2017-89657-P.

**Acknowledgments:** Our special thanks to Nigel Walkington for his extremely in-depth English revision; Enrique Fernández Seguí for his exceptional thorough work, formatting the text; some managers of Local Action Groups for their information/knowledge shared about the projects; the Andalusian Regional Government for the given data. Finally, Ernest Reig, Francisco Goerlich and Ignacio Cantarino, for the support in the establishment of rural typologies.

**Conflicts of Interest:** The authors declare no conflict of interest.

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
