# Peer review of "Entrepreneurs and Territorial Diversity: Success and Failure in Andalusia 2007–2015"

_land, doi:10.3390/land9080262_

Round 1

Reviewer 1 Report

The paper has the merit to shed light on “failed” projects within Leader programs. It provides an interesting territorial interpretation of the reasons why some (Andalusian) promoters despite had begun the LEADER grant application procedure, a grant was never ultimately obtained.

Minor revisions:

2 - verify "your"

151 - verify "SIG"

198 - verify brackets

202 - explain or link to table 1 "regions 6-7"

203 - explain or link to table 1 "region 5"

207 - explain or link to table 1 "category 7"

212 - explain or link to table 1 "category 7"

notes 1; 2; 3 - link to table 2 or write in full

351 (table 4) - add "distribution of"

536 - verify sentence

628 - verify agro-industrial (space)

861 - add a "g" (Belliggiano)

Reviewer 2 Report

The paper provides an interesting analysis of the LEADER measure implementation in Andalusia, comparing „successful“ and „failed“ project applications over the period of 2007-2015. As most studies on rural development and LEADER implementation start from a favourable view on the usefulness of LEADER for local development and rural development, in general, a critical survey of the amount and background of failure is particularly appreciated. The paper reveals the different access of various groups of project promoters and social groups to LEADER projects and the influence of spatial factors and location on the level of policy uptake. This implies that the objectives of LEADER to increase participation of social groups (like young people and women) and the support in remote parts of rural areas can only be partly achieved. As a result the LEADER scheme (and the complete rural development programme) are esteemed as hardly effective in halting large-scale depopulation processes in remote rural regions (of Andalusia).

The analysis for the paper makes use of a very detailed data base of all project applications for LEADER projects in the 2007-2013 period including 12,855 grant applications. This is an exceptional base and enables authors to address „failed“ projects in a very high detail. In comparison to other countries this is a much better resource base, as in many other countries only accepted project proposals are reported further and can be analysed. The only issue about this clear attribution of success and failure i the lack of qualitative information about the contents and details of both failed and successful projects. For both groups very different trajectories (and resubmission, redrafting and, on the other hand, implementation with little effect, termination of projects, no impact etc.) might also be observable. It might be helpful for readers to learn of such concerns on the background of the analysis carried out. Nevertheless, the current study seems a very valuable approach to shed light on the application process and different access and success level for different promoter groups.

I’d like to share some concerns on this draft of the paper which might be addressed in a (minor) revision of this paper. These address in particular:

  • A strong focus on Spanish literature, which is, of course, appropriate for analysing the case of Anadalusia, but more reference to international literature in the introductory presentation and the discussion section might increase the relevance of the paper for other contexts across Europe. This implies for example aspects of LEADER distribution across Europe (see SEGIRA study, with distribution of EU funds, ECORYS 2010; and distrubtion of Pillar 2 funds, see Shucksmith et al. 2005), studies on mitigating action toward depopulation trends of remote rural regions as a specific challenge (see e.g. ESPON 2020 study ESCAPE), research on the integration of young people and/or women in rural development (e.g. Shucksmith 2010, Jentsch and Shucksmith 2004), and on the discussions to update the regional classification system by OECD and EU (see e.g. the focus on grid data integration in Fadic et al. 2019). Moreover, literature on local strategies and critical assessment of LEADER application should be referred to uincreasingly (see e.g. Dax et al. 2016; reference 70 already addressed; Granberg et al. 2015; Papadopoulou et al. 2011, and others).
  • The other important point for revision is to increase clarity and legibility of results. The presentation of findings is rather lengthy (with long lists of results in seven tables and seven maps). In particular, in some parts of the text groups of promoters are addressed which are, however, not shown in the tables (e.g. there could be bold lines between the various groups to indicate them), and sometimes explanation in the text obviously refers to the groups average (which is not shown in the table), sometimes to sub-groups (taken from the tables). Moreover, it could be reconsidered if so many detailed figures actually have to be reiterated in the text or if it would not be better to focus on the messages (of comparison etc.). Moreover, the level of accuracy is diverse, sometimes rounding off figures, sometimes taking the exact value from the tables. A shorter presentation of these details would be appreciated.
  • There might also be some caveat on the reliabilty of assessment data. Jobs created and consolidated jobs are not always very reliable measurements in other Member States and some cautionary remarks might be appropriate. This might involve some bias which we should be aware of when carrying out the assessment.
  • The comparison to the EU context should be increased in the discussion. For example, the reduction of LEADER funds in Andalucia for period 2007-2013 in comparison to the previous period is reported whereas this period marked the „mainstreaming“ of LEADER into Rural Development Programmes (RDP) and an overall 3-4 fold extension of LEADER funds due to the requirement to spend at least 5% of EAFRD budget of RDP on LEADER measures. Obviously, this had no implications for Andalucia since the level of LEADER funding was already before that much higher. It might be relevant to address also the report and concern by the European Court of Auditors on LEADER implementation issues at the start of the mainstreaming period, and addressing administrative issues (European Court of Auditors 2010).
  • In the discussion section the reference to a limited use of the innovation potential of rural municipalities is highlighted. Also here more integration of qualitative discussions and literature might be relevant (e.g. OECD 2014, da Rosa Pires et al. 2014 etc.).
  • At last, the strong focus on women entrepreneurs in the last paragraph of the discussion (and the paper) is valid, but comes rather late. It should be seen as priority of regional strategy (or requirement to shift activities) throughout the paper. Simlarly depopulation is addressed here at the end. Both aspects might seem obvious in the regional context but a clearer focus on these, earlier on in the paper would be very useful.
  • (a last minor issue: please check the last sentence: it is too long and carries a series of messages which would be better realized if separated or explained step by step).

To sum up, the paper is a very rich account of the dependance of LEADER implementation on the diverse groups of actors (with its different roles) and the geography of the application where remote contexts are often adverse to policy impplementation. Consequences on future programmes and adaptation in regional action are very interesting.

References:

da Rosa Pires, A., Pertoldi, M., Edwards, J. and Hegyi, F.B. (2014) Smart specialization and innovation in rural areas. S3 Policy Brief Series No. 09/2014. Sevilla: European Commission, JRC-IPTS.

ECORYS (2010) Study on Employment, Growth and Innovation in Rural Areas (SEGIRA). Main report. commissioned by European Commission, Directorate-General for Agriculture and Rural Development, Brussels.

Dax, T., Strahl, W., Kirwan, J. and Maye, D. (2016) The Leader programme 2007-2013: Enabling or disabling social innovation and neo-endogenous development? Insights from Austria and Ireland, in: European Urban and Regional Studies 23(1), 56-68. DOI: 10.1177/0969776413490425.

European Court of Auditors (2010) Implementation of the  LEADER Approach for Rural Development. Special Report No 5/2010. Luxembourg.

Fadic, M., Garcilazo, J.E., Moreno Monroy, A. and Veneri, P. (2019) Classifying small (TL3) regions based on metropolitan population, low density and remoteness. OECD Regional Development Working Papers 2019/06, Paris. https://dx.doi.org/10.1787/b902cc00-en

Granberg, L., Andersson, K. and Kovách, I. (2015): Introduction: LEADER as an Experiment in Grass-Roots Democracy, in L. Granberg, K. Andersson, and I. Kovách (eds), Evaluating the European Approach to Rural Development, Grass-roots Experiences of the LEADER Programme. Perspectives on Rural Policy and Planning. Farnham: Ashgate, 1-12.

Jentsch B. and Shucksmith M. (eds), Young People in Rural Areas of Europe,

Ashgate, 2004.

OECD (ed.) (2014) Innovation and Modernising the Rural Economy, OECD publishing, Paris.  https://doi.org/10.1787/9789264205390-en.

Papadopoulou, E., Hasanagas, N. and Harvey, D. (2011) Analysis of rural development policy networks in Greece: Is LEADER really different? Land Use Policy 28 (4), 663-673. http://dx.doi.org/10.1016/j.landusepol.2010.11.005

Shucksmith, M., Thomson, K.J., and Roberts, D. (2005), CAP and the Regions: The Territorial Impact of Common Agricultural Policy, CABI Publishing, Wallingford (UK). (ISBN 0 85199 055 X).

Shucksmith, M. (2010) How to promote the role of youth in rural areas of Europe? Report for European Parliament, Directorate General for Internal Policies, Policy Department B: Structural and Cohesion Policies. Brussels.
